# An Avirulent Largemouth Bass Birnavirus Vaccine Candidate Protects Largemouth Bass against Birnavirus Infection

**DOI:** 10.3390/vaccines11121740

**Published:** 2023-11-22

**Authors:** Xiaozhe Fu, Mingju Luo, Qiang Lin, Hongru Liang, Yinjie Niu, Xia Luo, Baofu Ma, Ningqiu Li

**Affiliations:** Pearl River Fishery Research Institute, Chinese Academy of Fishery Sciences, Key Laboratory of Fishery Drug Development, Ministry of Agriculture and Rural Affairs, Guangdong Provincial Key Laboratory of Aquatic Animal Immune Technology and Green Breeding, Guangzhou 510380, China; fuxiaozhe@prfri.ac.cn (X.F.); 18893107156@163.com (M.L.); linq@prfri.ac.cn (Q.L.); hrliang@prfri.ac.cn (H.L.); nyj@prfri.ac.cn (Y.N.); luoxia@prfri.ac.cn (X.L.); mabf@prfri.ac.cn (B.M.)

**Keywords:** birnavirus, LBBV-GDSS-20180701, largemouth bass, live vaccine

## Abstract

Background: Largemouth bass birnavirus (LBBV) disease outbreaks in largemouth bass fingerlings lead to high mortality in China. Therefore, the development of immersion immunization strategies is paramount. Methods: An avirulent LBBV strain was screened using a fish challenge assay. The proliferation dynamics of the avirulent strain were determined in vitro and in vivo. The efficacy of the avirulent vaccine was evaluated using immune gene expression, viral load, and a virus challenge, and the safety was also assessed using a reversion to virulence test. Results: An avirulent virus strain, designated as largemouth bass birnavirus Guangdong Sanshui (LBBV-GDSS-20180701), was selected from five fish birnavirus isolates. The proliferation peak titer was 109.01 TCID50/mL at 24 hpi in CPB cells and the peak viral load was 2.5 × 10^4^ copies/mg at 4 dpi in the head kidneys and spleens of largemouth bass. The largemouth bass that were immersed within an avirulent vaccine or injected with an inactivated vaccine were protected from the virulent LBBV challenge with a relative percent survival (RPS) of 75% or 42.9%, respectively. The expression levels of IL-12, MHCI, MHCII, CD8, CD4, and IgM in the avirulent group were significantly upregulated at a partial time point compared to the inactivated vaccine group. Moreover, the viral load in the avirulent vaccine group was significantly lower than those in the inactivated vaccine group and control group using real-time PCR. Conclusions: LBBV-GDSS-20180701 is a potential live vaccine candidate against LBBV disease.

## 1. Introduction

The largemouth bass, *Micropterus salmoides*, has been an important freshwater cultured fish with more than 800,000 tons of production in 2022 in China [1]. However, virus diseases are becoming the greatest limiting factor in largemouth bass cultivation. *Siniperca chuatsi* rhabdovirus and largemouth bass ranavirus, which lead to high mortality, have been reported in largemouth bass cultured in China [2,3]. In recent years, outbreaks of epidemic LBBV disease have occurred in larval fish 2–6 cm in length, causing high mortality in China [4]. Clinical signs of LBBV infection include body color darkening, lethargy, irregular swimming behavior, punctate hemorrhages on the liver, and yellow glutinous fluid in the intestine [4].

LBBV is a kind of non-enveloped bi-segmented double-stranded RNA virus, and it belongs to the *Birnaviridae* family [4,5]. The *Birnaviridae* family has seven genera, including *Aquabirnavirus*, *Blosnavirus*, *Avibirnavirus*, *Dronavirus*, *Entomobirnavirus*, *Telnavirus*, and *Ronavirus*. The species of *Aquabirnavirus*, *Telnavirus*, *Blosnavirus*, and *Ronavirus* infect aquatic organisms [6]. The *Aquabirnavirus* species infects fish, crustaceans, and mollusks, and is an infectious pancreatic necrosis virus (IPNV) [5]. Our previous study found that LBBV shared a 45.5% sequence identity with IPNV but a 98.7% sequence identity with *Lates calcarifer* birnavirus (LCBV) based on the RdRp protein sequences, suggesting that LBBV and LCBV belong to a new genus [4].

Because of the absence of effective therapeutic methods, vaccination is considered one of the most viable strategies to control viral diseases. It has been reported that IPNV, the prototype virus of the Birnaviridae family, is very contagious and destructive to rainbow trout fingerlings, causing up to 70% mortality in hatchery stocks [7,8]. Our epidemiological investigation showed that juvenile largemouth bass were susceptible to LBBV, and mortality was nearly 100% [4]. Therefore, the development and improvement of massive immunization strategies are paramount. A live vaccine has better application prospects for LBBV protection because of the convenient immersion administration. However, there have been no live vaccines for the largemouth bass birnavirus until now.

In this paper, an avirulent birnavirus strain designated as largemouth bass birnavirus Guangdong Sanshui (LBBV-GDSS-20180701) was chosen from five LBBV strains. The immersion protective efficacy for largemouth bass against the virulent virus challenge was assessed and the risk of virulence reversion by passages of a vaccine candidate in vivo was evaluated. The results indicate that LBBV-GDSS-20180701 is a potential live vaccine candidate for largemouth bass against LBBV disease.

## 2. Materials and Methods

### 2.1. Fish, Cell Line, and Viruses

Largemouth bass 3–4 cm in length were purchased from farm in Guangdong, cultured in the recirculating aquaculture system, and fed daily with commercial feeds. The antibody against LBBV was negative according to a neutralization test, and fish were acclimated to laboratory conditions (28~30 °C) for 2 weeks before the experiment.

Chinese perch brain (CPB) cell line was constructed in our lab [9]. CPB cells were cultured in Leibovitz’s L15 medium (Labgic Technology Co., Ltd., Hefei, China) containing 8% fetal bovine serum (FBS, ExCell Bio. Co., Ltd., Shanghai, China) at 28 °C.

Five LBBV strains, composed of LBBV-GDQY-20170902, GDQY-20170701, LBBV-HNHY-20170401, LBBV-GDQY-20170901, and LBBV-GDSS-20180701, were isolated and stored in our lab. CPB cells grown in an L-15 medium (supplemented with 2% FBS) were inoculated with LBBV, and supernatants were stored at −80 °C until use.

### 2.2. Total RNA Extraction and Reverse Transcription

Total RNAs were extracted with Direct-zol RNA Miniprep kit (ZYMO RESEARCH, Irvine, CA, USA) based on the manufacturer’s protocols. Then, reverse transcription was performed with RevertAid™ First Strand cDNA Synthesis Kit (Vilnius, LT-02241, Lithuania), and cDNA was stored at −20 °C until used.

### 2.3. Screening of the Avirulent LBBV Strain

#### 2.3.1. Viral Titer Determination in CPB Cells

Five LBBV isolates multiplied in CPB cells. The viral titers were determined using TCID_50_ assay according to the method of Reed and Muench [10]. CPEs were observed and recorded for 10 consecutive days using the viral titer calculation.

#### 2.3.2. Determination of Virulence of Different Isolates with Challenge Experiment

A challenge experiment was designed with six groups, including one control group and five experimental groups (30 fish per group). The experimental groups were injected intraperitoneally (I.P.) with different LBBV strains at a dose of 5 × 10^4.0^ TCID_50_ in 0.05 mL volume solution per fish, and fish in control group were injected with 0.75% NaCl in 0.05 mL volume solution. Mortality was recorded for 10 days after the challenge.

According to the mortality rates, the strongest virulent strain and the avirulent strain were selected to further verify the virulence. Fish were intraperitoneally injected with 0.05 mL at a dose of 5 × 10^6.0^ TCID_50_ per fish. Others were treated the same as above.

### 2.4. Dynamics of the Avirulent Strain In Vitro and In Vivo

**In vitro:** Confluent monolayers of CPB cells were inoculated with avirulent LBBV strain at a multiplicity of infection (MOI) of 0.01. The viral titers were calculated using a TCID_50_ assay at 6, 12, 24, 36, 48, and 60 h post infection (hpi).

**In vivo:** Thirty-five healthy largemouth bass were injected intraperitoneally at a dose of 10^7.5^ TCID_50_ per fish. Head kidneys and spleens from five fish were sampled at 1, 2, 3, 4, 5, 6, and 7 days post infection (dpi). Head kidneys and spleens were weighed and the total RNAs were isolated and transcribed as described in Section 2.2. Viral copy numbers were determined using a real-time PCR with specific primers (LBBV-F, LBBV-R) and probes (Table 1). The qPCR assay was performed with the Real-time Detection System (Applied Biosystems) using the Premix Ex TaqTM Kit (Takara) according to the manufacturers’ instructions.

### 2.5. Immune Protection Evaluation

#### 2.5.1. Vaccine Preparation

Avirulent vaccine and inactivated vaccine were prepared. Briefly, 80–90% of confluent CPB cells were infected with LBBV-GDSS-20180701 (avirulent strain) or LBBV-GDQY-20170902 (virulent strain) at an MOI = 0.01. The cells and medium were harvested and then disrupted by 3 frozen–thaw cycles when 70–80% of cells showed cytopathic effects (CPEs). The virus suspension was centrifuged at 7500× *g* for 20 min at 4 °C and the viral supernatants were stored at −80 °C until use. The viral titer was calculated in the same way as in Section 2.3.1.

For virus inactivation, LBBV-GDQY-20170902 viral supernatants were inactivated with formalin at the final concentration of 0.1% at 37 °C with continuous stirring at 80 rpm for 48 h. Then, to the suspension was added 0.2% sodium metabisulfite for neutralizing the formalin. Finally, the inactivated virus and MontanideTM IMS1312 (Seppic) were mixed at a ratio of 1:1 (W/W) under moderate agitation for 5 min, using a magnetic stirrer.

#### 2.5.2. Evaluation of Antigenicity

The antigenicity evaluation experiment set up four groups: the avirulent vaccine group, the inactivated vaccine group, the immersion control group, and the injection control group (30 fish per group). For avirulent vaccine group, fish were immersed in 5 L of avirulent vaccine at a concentration of 10^6.0^ TCID_50/mL_, and for the immersion control group, fish were immersed in 5 L of water. For the inactivated vaccine group, fish were intraperitoneally injected with 0.05 mL of inactivated vaccine at a dose of 2.5 × 10^7.42^ TCID_50_ per fish, and for the injection control group, fish were injected with 0.05 mL of 0.75% NaCl. On the 21st dpv, fish were challenged by I.P. with virulent strain LBBV-GDQY-20170902 at a dose of 5 × 10^7.42^ TCID_50_ per fish. The mortality was recorded for 7 days post challenge. The spleen and kidney tissues from dead fish were sampled for qPCR analysis. Relative percentage survival (RPS) was calculated as follows: RPS = (1 − [Mortality of vaccinated fish/Mortality of unvaccinated control fish]) × 100%.

#### 2.5.3. Determining LBBV Load of Dead Fish with Real-Time PCR

At 1 d, 2 d, and 3 d post challenge, three largemouth bass from each group were sacrificed, and then spleens and head kidneys were sampled in time for LBBV load experiment in order to assess the vaccine’s effect on virus clearance. Head kidneys and spleens were weighed and the total RNAs were isolated and transcribed as described above. The LBBV load was detected by real-time PCR in the same way as in Section 2.4.

#### 2.5.4. Transcription of Immune-Related Genes Post Vaccination

The immune gene transcription levels in fish from different groups were evaluated using RT-qPCR assays. Total RNAs from the spleens of three fish in each treatment group were isolated on 0, 1, 2, 3, 4, 5, 6, and 7 dpv for detection of IL-12, MHCI, MHCII, CD8, and CD4 gene expression, or from head kidneys on 0, 7, 14, and 21 dpv for detection of IgM gene expression. The RNA concentration was adjusted to 5 μg for cDNA reverse transcription. The qPCR was performed in an ABI 7500 Real-time Detection System according to the Maxima SYBR Green/ROX qPCR Master Mix (Fermentas, CAN) Kit manipulation instructions. The immune gene expression level was calculated with the formula F = 2^−ΔΔCt^, ΔΔCt = (Ct_, target gene_ − Ct_, reference gene_) vaccine − (Ct_, target gene_ − Ct_, reference gene_) control. The 18SrDNA gene was used as an internal control. All data were expressed as means ± SD. The primers used in this paper are listed in Table 1.

### 2.6. Safety Evaluation of Avirulent Vaccine

Reversion to virulence test was performed to evaluate the safety of avirulent vaccine. Largemouth bass were intraperitoneally injected with 0.05 mL of avirulent vaccine at a dose of 10^7.5^ TCID_50_ per fish or L-15 medium as a control. Fifteen fish were used for every passage test, and five fish were randomly sampled at 4 dpi for the next passage injection. A part of the sampled spleen and kidney were subjected to virus detection using cell culture on CPB and RT-PCR constructed in our lab [11]. Then, the other parts of the sampled organs were homogenized on ice with L-15 medium. The homogenate was centrifuged at 5000 rpm at 4 °C for 15 min, then filtered through a 0.22 μm pore size membrane filter, and supernatants were used for the next passage injection. Subsequently, fifteen largemouth bass were injected with 0.05 mL viral supernatant and reared in the tanks. The injection step was repeated four times for the avirulent vaccine strain passage in largemouth bass. At the last passage, five fish were sampled for LBBV detection by RT-PCR and cell culture, and the remaining ten fish were reared for 21 days for abnormality and mortality observation.

### 2.7. Statistical Analysis

The relative immune-related gene expression and LBBV loads in the different groups were analyzed using one-way ANOVA (SPSS, Chicago, IL, USA). Significance tests were determined if there was a statistically significant difference. The χ^2^ test was used to evaluate LBBV loads in different tissues and the protective efficacies of the different groups. The significance level was set at *p* < 0.05.

## 3. Results

### 3.1. Screening of the Avirulent Largemouth Bass Birnavirus Strain

The virulence levels of five LBBV isolates were assessed. The results indicated that the mortality rates of LBBV-GDQY-20170902, GDQY-20170701, LBBV-HNHY-20170401, LBBV-GDQY-20170901, and LBBV-GDSS-20180701 were 26.7%, 20%, 13.3%, 6.7%, and 0%, respectively (Figure 1a). To further verify the virulent and avirulent strains, the challenge experiment was carried out again for GDQY-20170902 and GDSS-20180701 using the increased challenge dose. As shown in Figure 1b, the mortality rates of LBBV-GDQY-20170902 and LBBV-GDSS-20180701 were 73% and 0%, respectively. All of the above illustrated that LBBV-GDQY-20170902 was a virulent strain and LBBV-GDSS-20180701 was an avirulent strain.

### 3.2. Dynamics of the Avirulent Strain In Vitro and In Vivo

The growth curve of the avirulent strain in the CPB cells was determined using a TCID_50_ assay. As shown in Figure 2a, the viral titer sharply increased from 12 to 24 hpi and then decreased from 24 to 48 hpi. The peak titer was 10^9.01^ TCID_50_/mL at 24 hpi.

The proliferation of the avirulent strain in the largemouth bass was measured using qPCR (Figure 2b). The results showed that the LBBV copy numbers increased slowly before 3 dpi, then sharply reached a peak at 4 dpi, where the viral load was 2.5 × 10^4^ copies/mg, and then declined rapidly, until undetection at 7 dpi.

### 3.3. The Kinetics of Immune-Related Gene Expression in Different Vaccine Groups

The gene expression levels of IL-12, MHCI, MHCII, CD8, and CD4 in spleens on 0, 1, 2, 3, 4, 5, 6, and 7 dpv and IgM in head kidneys on 0, 7, 14, and 21 dpv of three fish in each group were examined using qRT-PCR. The kinetics of the above gene expressions are shown in Figure 3. For the group vaccinated with the avirulent vaccine, the MHCI and CD8 gene expressions exhibited significant up-regulation with respective 18.3 and 8.3-fold increases at 1 dpv compared to the control group, and then they rapidly decreased. But for the group vaccinated with the inactivated vaccine, the MHCI gene expression showed no significant changes. As for the MHCII gene, a different expression pattern was observed. For the group vaccinated with the avirulent vaccine, MHCII expression was significantly increased at all time points examined and peaked at 4 dpv, with a 6.1-fold increase. For the group vaccinated with the inactivated vaccine, the MHCII expression increased at 7 dpv with a 5.1-fold increase and had no significant change before 7 dpv. Additionally, the CD4+ gene expression in the avirulent group also peaked at 1 dpv and 6 dpv with 16.3-fold and 4.1-fold increases compared to the control group. However, the inactivated group’s CD4 gene expression was mildly up-regulated and peaked at 7 dpv with a 3.3-fold increase. The IL-12 gene expression in the avirulent group up-regulated rapidly and reached a peak at 1 dpv with a 19.2-fold increase compared to the control group and then peaked again with 11.9-fold increase at 4 dpv. However, for the inactivated group, the IL-12 gene expression showed no significant change. At all the time points examined, IgM expression was significantly up-regulated compared to the control group and reached a peak with a 64-fold increase at 21 dpv. For the inactivated group, IgM expression was also up-regulated and peaked at 21 dpv with an 8.6-fold increase.

### 3.4. Immune Protection of Different Vaccines against the LBBV Challenge

Figure 4a shows the cumulative mortality post-challenge with LBBV-GDQY-20170902 at 21 dpv. Dead fish from the injection control group were recorded from the 2nd to the 5th day post challenge, and the cumulative mortality was 84%. Dead fish from the immersion control group were also observed from the 2nd to the 5th day post challenge, and the cumulative mortality was 80%. However, dead fish from the avirulent vaccine group were observed from the 3rd to the 4th day post challenge, and the cumulative mortality was 20%. For the inactivated vaccine group, dead fish were recorded from the 2nd to the 4th day post challenge, and the cumulative mortality was 48%. As we can see from Figure 4b, the protective efficacy of the avirulent vaccine group was the highest, and the RPS value was 75%. The RPS value of the inactivated vaccine was 42.9%. Furthermore, the dead fish showed typical LBBV infection symptoms, and no other pathogens except LBBV were detected in the dead fish.

### 3.5. LBBV Loads of the Dead Fish in Different Groups

LBBV loads were detected by real-time PCR in the head kidneys and spleens of largemouth bass. Figure 5 shows that the viral loads in the head kidneys and spleens from different groups decreased with death time. The viral loads in the head kidneys from the avirulent group, inactivated group, immersion control group, and injection control group post challenge were 831~2977, 995~4012, 12,921~85,980, and 9921~82,980 copies/mg, respectively. The viral loads in the spleens from the avirulent group, inactivated group, immersion control group, and injection control group post-challenge were 4786~11,981, 20,480~21,716, 23,800~32,973, and 21,800~42,973 copies/mg, respectively. Interestingly, for the control group, the viral load in the head kidney was higher than that in the spleen, but for the vaccinated group, the viral load in the spleen was higher.

### 3.6. Virulence Reversion Test of the Avirulent Vaccine

Reversion to virulence is the first key for avirulent vaccine safety testing. In this paper, the kidney and spleen homogenates prepared from largemouth bass injected with the avirulent vaccine were passaged in largemouth bass up to five times, and viral genomic RNA was detected by RT-PCR. As shown in Figure 6, PCR analysis revealed that a 339 bp amplified fragment was observed from the first fish-to-fish passage. However, no PCR product was amplified, and no live virus was isolated from passage 2 to passage 5. Furthermore, the fish that were administrated avirulent vaccine passages 1 to 5 and those in the control group showed no mortalities or abnormalities. The results illustrated no virulence reversion sign with the vaccine strain passages in the fish.

## 4. Discussion

Vaccination is an effective prevention strategy to control viral diseases in fish. Although fish can be administrated vaccines by injection (i.p. or i.m.) fish to fish, immersion or oral administration (mass delivery) methods, the best delivery method is immersion [12,13]. Compared to the other kinds of fish vaccines, live-attenuated vaccines have many advantages because they are not only able to mimic natural viral infections inducing the host immune response, but can also be delivered via immersion [14,15,16]. LBBV is an emerging fish birnavirus and has caused high mortality in largemouth bass fingerlings [4]. Thus, developing a live vaccine is ideal for protecting largemouth bass against LBBV infection. IPNV is the prototype virus of Aquabirnavirus [17]. Previous studies showed that IPNV causes salmonid juvenile mortality worldwide, especially for salmonid eggs and fingerlings [5,6]. Until now, several injectable vaccines against IPNV based on the inactivated virus or recombinantly viral peptides have been used in some countries, but the real protection levels in field conditions are variable and much lower than those obtained in experimental trials [13,16]. Scientists discovered that these vaccines could stimulate the fish humoral immune response but do not protect efficiently against IPNV, suggesting that non-specific cytotoxic cells (NCC) of the innate immune response may be necessary for IPNV prevention [18,19]. Thus, several live vaccines, such as provirus and live vector vaccines, that activate the innate immune and humoral immune response have been developed, and the live vector vaccine had a high protection rate against IPNV [18,19]. In this study, one naturally avirulent LBBV strain was successfully screened and its viral titer in CPB cells reached over 10^9.0^ TCID_50_/_mL_, suggesting that it was suitable to produce an avirulent vaccine. Then, the vaccine efficacy of the avirulent LBBV strain was tested, and it showed an RPS of 75%, which is higher than the RPS of 42.9% of the inactivated vaccine. This proved that the avirulent LBBV strain could provide stronger protection to largemouth bass against LBBV.

A previous study showed the surviving fish post IPNV infection were asymptomatic carriers; therefore, a good IPNV vaccine should eliminate residual IPNV after the challenge. In this study, the LBBV loads in the head kidneys and spleens of the avirulent vaccine group were significantly decreased post challenge compared to the control group and inactivated group. Especially, the viral loads in the spleens of the avirulent vaccine group continuously and significantly declined at 1, 2, and 3 dpi, whereas the viral loads of the inactivated vaccine group slightly declined at 1 and 2 dpi, and no significant difference from the control group was observed at 3 dpi. Thus, it was possible for the avirulent vaccine to eliminate residual LBBV.

It is reported that humoral responses and cellular responses are critical for virus prevention and clearance. The cellular responses clear virions in the infected host cells through the CTL response, and the humoral responses eliminate extracellular virions through the antibody responses [20]. In this study, the gene expression levels of MHCI, MHCII, CD8, CD4, IL-12, and IgM in the avirulent group were up-regulated post vaccination compared to the inactivated vaccine group, suggesting that the avirulent vaccine could induce humoral and cellular immune responses. IL-12 is a kind of cytokine used to promote cytotoxicity of cytotoxic T and NK cells and boost the differentiation of the T helper (Th) 1 cell [21]. Th 1 cells play a very important role in the elimination of viral infections [22]. Thus, we inferred that the avirulent vaccine maybe activate Th1 cell immunity.

However, reversion to virulence is a key risk for the live vaccine. So far, limited live attenuated vaccines have been approved in aquaculture, including the modified live F. columnare vaccine for channel catfish in the USA [23], the attenuated KHV disease vaccine for koi carp in Israel, and the avirulent GCRV disease vaccine for grass carp in China [14]. In this study, the virulence reversion test was assessed using five fish-to-fish passages in largemouth bass, and a positive product was only observed from the first passage. Furthermore, no abnormalities or mortalities were recorded, verifying that the avirulent strain could invade but not revert to virulence. It was reported that the live vaccine’s safety was also verified by performing in vivo reversion back passage for the modified live *F. columnare* vaccine [24], and the live attenuated cyprinid herpesvirus-2 vaccine [25]. Therefore, it verified that the avirulent LBBV vaccine was safe in largemouth bass.

## 5. Conclusions

An avirulent virus strain LBBV-GDSS-20180701 was screened. We found that it has good immunogenicity for largemouth bass against virulent LBBV with IM vaccination. Thus, LBBV-GDSS-20180701 is a potential live vaccine candidate against LBBV disease.

## Figures and Tables

**Figure 1 vaccines-11-01740-f001:**
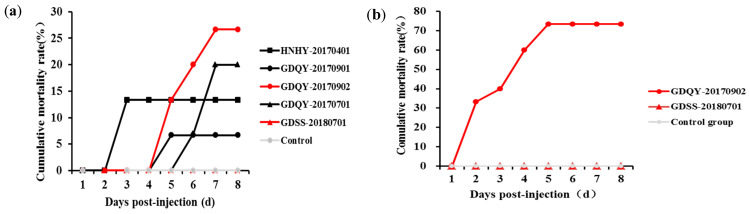
Cumulative mortality curves for different experimental groups injected with different LBBV strains. (**a**) At a dose of 5 × 10^4^ TCID_50_ per fish. (**b**) At a dose of 5 × 10^6^ TCID_50_ per fish.

**Figure 2 vaccines-11-01740-f002:**
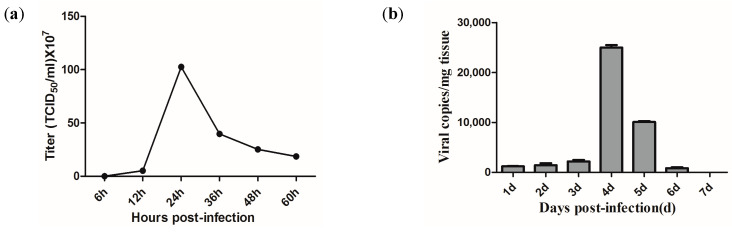
Proliferation dynamics of avirulent strain in CPB cells (**a**) and in largemouth bass (**b**).

**Figure 3 vaccines-11-01740-f003:**
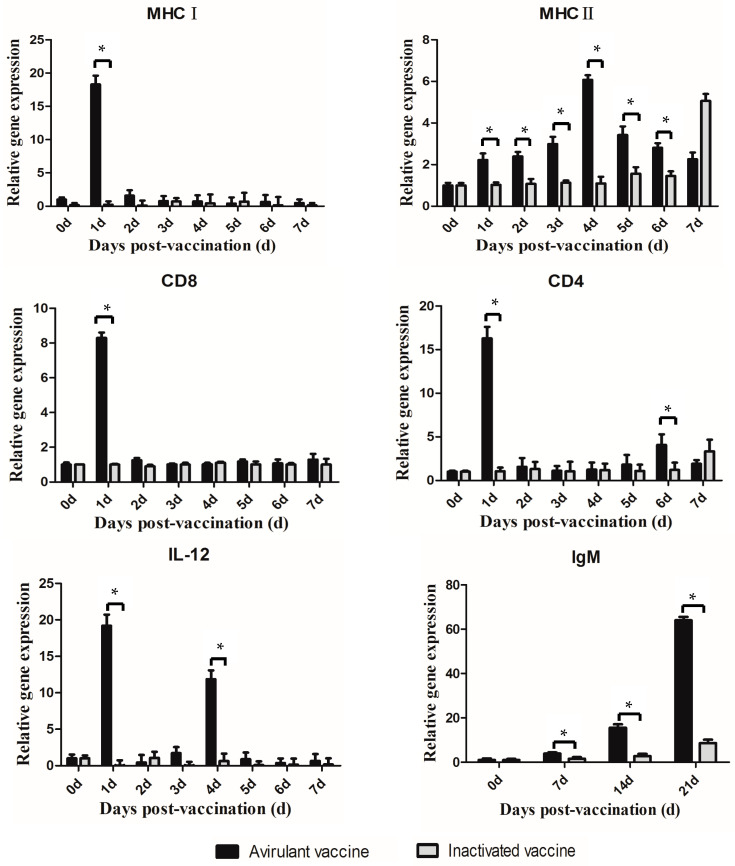
qRT-PCR analysis of the immune-related gene expressions in different vaccine groups. Total RNAs were extracted from the spleen tissues on 0 d, 1 d, 2 d, 3 d, 4 d, 5 d, 6 d, and 7 d for detection of IL-12, MHCI, MHCII, CD8 and CD4, and from the head kidney tissues on 0 d, 7 d, 14 d, and 21 d for IgM detection post vaccination for use in qRT-PCR. * Significant differences from the inactivated group. Data are presented as means ± SE (N = 3), *p* < 0.05.

**Figure 4 vaccines-11-01740-f004:**
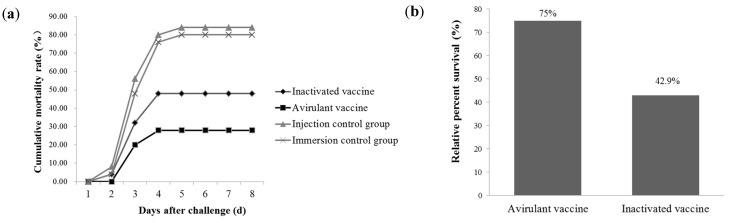
Immune protection of different vaccines against LBBV challenge. (**a**) Cumulative mortality curves for different groups after challenge with virulent LBBV. (**b**) Relative percent survival of different vaccine groups.

**Figure 5 vaccines-11-01740-f005:**
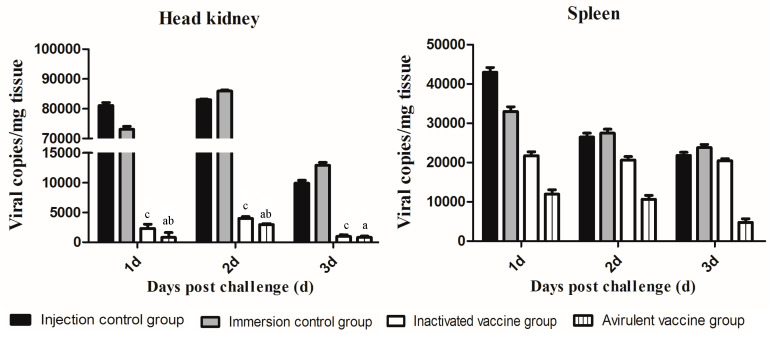
Detection of LBBV loads in largemouth bass from different groups. Fish from avirulent, inactivated, immersion control, and injection control groups at 1, 2, and 3 days post challenge were collected. Then, head kidneys and spleens were sampled immediately for LBBV load experiment. The viral copy number per gram of different tissues was determined by real-time PCR. ^a^ represents that avirulent vaccine group was significantly different from immersion control group (*p* < 0.05). ^b^ represents avirulent vaccine group was significantly different from inactivated vaccine group. ^c^ represents inactivated vaccine group was significantly different from injection control group.

**Figure 6 vaccines-11-01740-f006:**
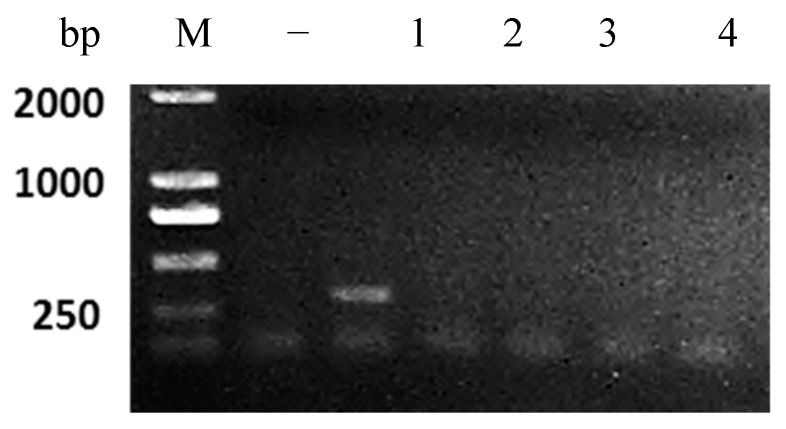
RT-PCR detection of avirulent vaccine strain in largemouth bass after serial propagations. M: DNA marker (DL2000); −: negative control; 1, 2, 3, 4, and 5: samples from 5 serial passages in largemouth bass vaccinated with avirulent strain. The size of amplified fragment is 339 bp.

**Table 1 vaccines-11-01740-t001:** The primers used in this study.

Gene Name	Accession NO.	Primer Name	Sequence(5′-3′)	Application
VP1	MW727623.1	LBBV-qF LBBV-qR Probe	AATCCAAAAACAACACGCTAAACA GCGCCTCATGATTGAGTCAAG (FAM)-ATGGGTTCAATCCCTTCAACGGCG-(Eclipse)	LBBV load
		LBBV-F LBBV-R	CCTGTCGTGCGGGCTCCTATT CTCTTTGTGGCGTTGGCTTCG	Virulent reversion test
IL-12bβ	XM_038708060.1	IL-12-F IL-12-R	TCTTCCATCCTTGTGGTCTTCC CAGTTCCAGGTCAAAGTGGTC	Gene expression
MHCIα	XM_038725867.1	MHCI-F MHCI-R	GTGGTTCAACGTCAACATCG ACCCAGACTTGTTCGGTGTC	
MHCIIα	XM_038711500.1 XM_038711494.1	MHCII-F MHCII-R	TCTACCCTGCAGAAGAAGCTCA CTCACTGGACGACCATTTTTAGTC	
CD8α	XM_046076130.1	CD8-F	GCATTTATAGCTGCGGTTTGC	
		CD8-R	GTTTGGCGGTGGTCCGTGTT	
CD4	XM_038711094.1 XM_038711102.1	CD4-F CD4-R	TGGTATCATCGTGGTAACTTCA AGCATCTTCTTCCTTCACTCCC	
IgM-H	MN871984.1	IgM-F IgM-R	TGGTGACCCTGACTTGCTATG GAGTCTGCTTCCTCGTCATCAAC	
18SrRNA	XR_005442393.1	18S-F 18S-R	GGACACGGAAAGGATTGACAG CGGAGTCTCGTTCGTTATCGG	

## Data Availability

Data is contained within the article.

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
