# Peer review of "An Avirulent Largemouth Bass Birnavirus Vaccine Candidate Protects Largemouth Bass against Birnavirus Infection"

_vaccines, 2023, doi:10.3390/vaccines11121740_

Round 1

Reviewer 1 Report

Comments and Suggestions for Authors

I believe that this paper is of great value in that it demonstrates the efficacy of a live attenuated vaccine against the important disease LBBV in largemouth bass, an important freshwater aquaculture fish in China. Furthermore, the virulence of the attenuated strain is extremely low, and we believe that it is at a level where there are no problems with its actual use at aquaculture sites. It would be of even greater value if the genetic differences between attenuated and highly virulent strains could be discussed, but this is a topic for another paper.

However, the following questions and improvements to the table regarding the primers used in the quantitative PCR for host immunity-related genes should be corrected as follows.

For primers in Table 1, for all primer sets, the gene ID or accession number of the source sequence should be provided so that the reader can see where it was designed. If there are several paralogs or subunits, it would be easier for the reader to understand if the reason why the molecule was chosen (e.g., because it has the highest expression level) is noted.

About IL12

Largemouth bass have two paralog molecules, IL12a and IL12b, as orthologs of mammalian IL12. IL12 is also a dimer of alpha and beta chains. The gene ID or accession number should be appended so that it is clear which molecule the primers are designed from. The gene name could be IL12a alpha, IL12a beta, IL12b alpha, or IL12b beta. In fact, the primer set seems to be designed from IL12b beta, so the gene name should be IL12b beta. Therefore, the primer name should be changed to the appropriate one. The two primer names are the same, IL-12-F, but the lower primer is a reverse primer and should be changed to the correct one.

About MHC I

MHC class I is a dimer of α and β chains. There are several paralogs of MHC class I in largemouth bass, both α- and β-chains. The primers seem to be designed from one of the α-chain, but it is necessary to specify the gene ID or accession number of which paralog molecule they are designed from. If the primers are common to multiple paralogs, this should also be stated. Gene and primer names should be changed accordingly.

About MHC II

MHC class II is a dimer of α- and β(γ?) chains. There are several paralogs for the α-chain β(γ?) chain of MHC class II in largemouth bass. The gene ID or accession number of which molecule it is designed from should be clearly indicated. If the primers are common to several paralogs, this should also be stated. The gene and primer names should be changed accordingly. I could not find the actual sequence of the primers in any of the Genbank registries. If the authors have designed primers in the sequences of new paralogs that authors have identified, authors should register the sequences in Genbank and state the accession numbers.

About CD8

Although the gene name is marked CD8+, the + is unnecessary. The + means "expressed" (e.g., CD8+ T cell means a T cell expressing CD4). Also, CD8 is a dimer of α and β chains. There is only one known type of CD8 in black bass, both α-chain and β-chain. Since the primers seem to be designed from the α-subunit, it is necessary to specify this. Accordingly, the gene and primer names need to be changed to appropriate ones.

About CD4

Although the gene name is marked CD4+, the + is unnecessary. The + means "expressed" (e.g., CD4+ T cell means a T cell expressing CD4). Although CD4 is monomeric, three paralogs (GeneID: 119897414, 119897410, 119897409) are known for sequences that appear to be orthologs of mammalian CD4 in largemouth bass. The primers should specify from which paralog the primers are designed, but I was unable to confirm the sequence of primers from any of the genes. However, the sequence of CD4-R was confirmed that 25 bases from the 5' end are located at the end of the 9th exon of the gene with GeneID:119897409, and the remaining 3 bases from the 3' end are located at the beginning of the 10th exon of this gene However, there was a mismatch at 14-15 bases from the 5' side of the primer. The primers are oriented backwards and no forward primer sequence could be found. If the authors have independently identified a new paralog and have designed primers from it, authors should register it with Genbank and state its accession number. The gene and primer names should be changed accordingly.

About IgM

IgM is a pentamer or hexamer consisting of heavy and light chains and sometimes J chains. Primers are designed from the heavy chain, so the gene and primer names need to be changed to appropriate ones so that they are recognizable.

About 18S

18S is not an appropriate gene name; it should be 18SrRNA or ssrRNA. There is also a mismatch with the 18SrRNA sequence registered in the fourth base from the 3' end of the reverse primer. If the authors designed the primers in the sequence of a new 18SrRNA variant that you identified yourself, arthors should register the sequence in Genbank and specify the accession number.

Author Response

Reviewer 1

I believe that this paper is of great value in that it demonstrates the efficacy of a live attenuated vaccine against the important disease LBBV in largemouth bass, an important freshwater aquaculture fish in China. Furthermore, the virulence of the attenuated strain is extremely low, and we believe that it is at a level where there are no problems with its actual use at aquaculture sites. It would be of even greater value if the genetic differences between attenuated and highly virulent strains could be discussed, but this is a topic for another paper. 

However, the following questions and improvements to the table regarding the primers used in the quantitative PCR for host immunity-related genes should be corrected as follows.

 For primers in Table 1, for all primer sets, the gene ID or accession number of the source sequence should be provided so that the reader can see where it was designed. If there are several paralogs or subunits, it would be easier for the reader to understand if the reason why the molecule was chosen (e.g., because it has the highest expression level) is noted.

Reply: The gene Accession NO. has been inserted into Table 1. The wrong primer names have been corrected and relavant name was also reviesed in figure 3 and manuscript.

About IL12 

Largemouth bass have two paralog molecules, IL12a and IL12b, as orthologs of mammalian IL12. IL12 is also a dimer of alpha and beta chains. The gene ID or accession number should be appended so that it is clear which molecule the primers are designed from. The gene name could be IL12a alpha, IL12a beta, IL12b alpha, or IL12b beta. In fact, the primer set seems to be designed from IL12b beta, so the gene name should be IL12b beta. Therefore, the primer name should be changed to the appropriate one. The two primer names are the same, IL-12-F, but the lower primer is a reverse primer and should be changed to the correct one.

Reply: Gene name has been revised to IL-12bβ, and gene Accession NO. was XM_038708060.1. Reverse primer was revised to IL-12-R.

 About MHC I 

MHC class I is a dimer of α and β chains. There are several paralogs of MHC class I in largemouth bass, both α- and β-chains. The primers seem to be designed from one of the α-chain, but it is necessary to specify the gene ID or accession number of which paralog molecule they are designed from. If the primers are common to multiple paralogs, this should also be stated. Gene and primer names should be changed accordingly.

Reply: Gene name has been revised to MHCâ… α, and gene Accession NO. was XM_038725867.1.

 About MHC II 

MHC class II is a dimer of α- and β(γ?) chains. There are several paralogs for the α-chain β(γ?) chain of MHC class II in largemouth bass. The gene ID or accession number of which molecule it is designed from should be clearly indicated. If the primers are common to several paralogs, this should also be stated. The gene and primer names should be changed accordingly. I could not find the actual sequence of the primers in any of the Genbank registries. If the authors have designed primers in the sequences of new paralogs that authors have identified, authors should register the sequences in Genbank and state the accession numbers 

Reply: We are very sorry to copy wrong primer sequences and we have corrected the primer sequenced and revised gene name to MHCâ…¡α. MHCâ…¡have two paralogs in largemouth bass in Gene bank, and primers could amplify the two gens, so we added gene Accession NO. as XM_038711500.1 and XM_038711494.1 in table 1.

About CD8 

Although the gene name is marked CD8+, the + is unnecessary. The + means "expressed" (e.g., CD8+ T cell means a T cell expressing CD4). Also, CD8 is a dimer of α and β chains. There is only one known type of CD8 in black bass, both α-chain and β-chain. Since the primers seem to be designed from the α-subunit, it is necessary to specify this. Accordingly, the gene and primer names need to be changed to appropriate ones. 

Reply: Gene name has been revised to CD8α, and gene Accession NO. was XM_046076130.1.

About CD4

Although the gene name is marked CD4+, the + is unnecessary. The + means "expressed" (e.g., CD4+ T cell means a T cell expressing CD4). Although CD4 is monomeric, three paralogs (GeneID: 119897414, 119897410, 119897409) are known for sequences that appear to be orthologs of mammalian CD4 in largemouth bass. The primers should specify from which paralog the primers are designed, but I was unable to confirm the sequence of primers from any of the genes. However, the sequence of CD4-R was confirmed that 25 bases from the 5' end are located at the end of the 9th exon of the gene with GeneID: 119897409, and the remaining 3 bases from the 3' end are located at the beginning of the 10th exon of this gene However, there was a mismatch at 14-15 bases from the 5' side of the primer. The primers are oriented backwards and no forward primer sequence could be found. If the authors have independently identified a new paralog and have designed primers from it, authors should register it with Genbank and state its accession number. The gene and primer names should be changed accordingly.

Reply: We are very sorry to copy wrong primer sequences and we have corrected the primer sequenced and revised gene name to CD4. Largemouth bass CD4 have three PREDICTED paralog genes in Genebank, in which Gene ID 119897414 have very high identity with Gene ID 119897410, but very low identity with Gene ID 119897409. So the listed primers could only amplify the Gene ID 119897414 and 119897410. We added gene Accession NO. XM_038711094.1 and XM_038711102.1 in table 1.

 About IgM 

IgM is a pentamer or hexamer consisting of heavy and light chains and sometimes J chains. Primers are designed from the heavy chain, so the gene and primer names need to be changed to appropriate ones so that they are recognizable.

Reply: Gene name has been revised to IgM-H, and gene Accession NO. was MN871984.1

 About 18S 

18S is not an appropriate gene name; it should be 18SrRNA or ssrRNA. There is also a mismatch with the 18SrRNA sequence registered in the fourth base from the 3' end of the reverse primer. If the authors designed the primers in the sequence of a new 18SrRNA variant that you identified yourself, arthors should register the sequence in Genbank and specify the accession number.

Reply: We are very sorry to copy wrong primer sequences and we have corrected the primer sequenced and revised gene name to 18SrRNA. We added gene Accession NO. XR_005442393.1 in table 1.

Reviewer 2 Report

Comments and Suggestions for Authors

Largemouth bass birnavirus (LBBV) is an emerging virus in China that is beginning to make a significant impact on freshwater fish aquaculture. Preventive measures to control outbreaks of viral diseases are much needed. In here, the potential use of an attenuated strain of LBBV as a vaccine was explored. Overall, the experimental data support the authors claims on the safety and efficacy of the live vaccine in largemouth bass. However, one important issue have been overlooked in this work:, the anti-LBBV antibody production in vaccinated fish was not analyzed. Bearing in main the fundamental role of humoral response in protection against pathogens I consider this to be a flaw in this work.  I would encourage the authors to present some data on this regard.

Specific issues

- In Figure 2a, Y axis, TCID50 values are indicated as “x107”. In the text, however (line 201) a peak titer of 109,01 TCID50/ml (that is only “102”) was written. Fig 2b is kind of superfluous. Also, figure 2 caption could be more informative.

 - In Figure 3, MHC I, CD8+ and CD4+expressions showed a sharp peak at 1 day post-vaccination. Have this kind of rapid “on/off” response been reported before for other live virus vaccines in fish? 

 - In Discussion (304-305), the two references cited to claim that “live vaccines protect against IPNV” (#18 and #19)  actually do not show any data regarding the vaccine efficacy in vivo. The authors may want to reconsider to delete or rephrase that sentence.

 - To my understanding (fig 2b) the LBBV attenuated strain was cleared from the fish body after 7 days. Thus, the vaccinated fish will never become virus carriers, and therefore the safety of the vaccine is guaranteed. Is that correct? On this respect, were any vaccinated fish tested for LBBV RNA at later times (ie at 21 dpv, the time of the virus challenge)?

- Figure 6: this experiment shows detection of LBBV RNA only in the first virus passage in fish. (The technique would be better called RT-PCR than PCR). I believe that the presence of viral RNA in fish could be better analyzed by the more sensitive real-time RT-PCR instead of standard PCR. Also, samples from the subsequent passages could have been tested on cell culture for the appearance of cytopathic effect to make sure that those fish were virus-free.

Author Response

Reviewer 2

Largemouth bass birnavirus (LBBV) is an emerging virus in China that is beginning to make a significant impact on freshwater fish aquaculture. Preventive measures to control outbreaks of viral diseases are much needed. In here, the potential use of an attenuated strain of LBBV as a vaccine was explored. Overall, the experimental data support the authors claims on the safety and efficacy of the live vaccine in largemouth bass. However, one important issue have been overlooked in this work: the anti-LBBV antibody production in vaccinated fish was not analyzed. Bearing in main the fundamental role of humoral response in protection against pathogens I consider this to be a flaw in this work.  I would encourage the authors to present some data on this regard.

Reply: Because we have not the monoclonal antibody against largemouth bass IgM, so we did not test anti-LBBV antibody production. But we detected IgM mRNA expression post vaccination, and in the later study we will test antibody production when we obtaint the monoclonal antibody against largemouth bass IgM.

Specific issues

- In Figure 2a, Y axis, TCID50 values are indicated as “x107”. In the text, however (line 201) a peak titer of 109,01 TCID50/ml (that is only “102”) was written. Fig 2b is kind of superfluous. Also, figure 2 caption could be more informative.

Reply: In order to show the obvious trend of virus proliferation in figure, TCID50 values of Y axis are indicated as “x107”, but not “x109”. As to Fig 2b, it is used for supplying the sampling time for vaccine safety evaluation. Figure 2 caption was revised to “Proliferation dynamics of avirulent strain in CPB cells (a) and in largemouth bass (b).”

 - In Figure 3, MHC I, CD8+ and CD4+expressions showed a sharp peak at 1 day post-vaccination. Have this kind of rapid “on/off” response been reported before for other live virus vaccines in fish? 

Reply: In the paper“Development of a live vector vaccine against infectious pancreatic necrosis virus in rainbow trout” (Shouhu Li et al., Aquaculture 524 (2020) 735275), CD4 and CD8 were continuously increased in the spleens between 3 and 15 dpv. In the another paper “Immune responses elicited in rainbow trout through the administration of infectious pancreatic necrosis virus-like particles” (S. Martinez-Alonso et al, Developmental and Comparative Immunology 36 (2012) 378–384), CD4 expression was detected, but not CD8. We speculated that the expression modle maybe relevant with virus proliferation dynamics in host or disease onset speed.

 - In Discussion (304-305), the two references cited to claim that “live vaccines protect against IPNV” (#18 and #19)  actually do not show any data regarding the vaccine efficacy in vivo. The authors may want to reconsider to delete or rephrase that sentence.

Reply: We have rephrased that sentence as follows: Thus, several live vaccines such as provirus and live vector vaccine, activating the innate immune and humoral immune has been developed and live vector vaccine had a high protection rate against IPNV [18, 19].

 - To my understanding (fig 2b) the LBBV attenuated strain was cleared from the fish body after 7 days. Thus, the vaccinated fish will never become virus carriers, and therefore the safety of the vaccine is guaranteed. Is that correct? On this respect, were any vaccinated fish tested for LBBV RNA at later times (ie at 21 dpv, the time of the virus challenge)?

Reply: Yes, the result indicates the LBBV avirulent vaccine is guaranteed. We did not detected LBBV at later time for safty experiment, but LBBV was detected after challenge for Immune protection experiment.

- Figure 6: this experiment shows detection of LBBV RNA only in the first virus passage in fish. (The technique would be better called RT-PCR than PCR). I believe that the presence of viral RNA in fish could be better analyzed by the more sensitive real-time RT-PCR instead of standard PCR. Also, samples from the subsequent passages could have been tested on cell culture for the appearance of cytopathic effect to make sure that those fish were virus-free.

Reply: We have corrected the PCR to RT-PCR. Cell culture is a gold standard for live virus seperation, but its sensitivity was lower than RT-PCR, so we referred to other papers to use RT-PCR for virulence reversion test.

Round 2

Reviewer 2 Report

Comments and Suggestions for Authors

I don´t feel satisfied with the authors responses to my questions and remarks. I understand that it is hard to fully address every single issue raised by one reviewer, but in this case the authors have not addressed most of the issues I posted in my original review.

This is the list of things I requested:

1.- Provide some data on fish antibody response to the vaccine > cannot be done (don´t have the anti-IgM antibody).

2.- Provide an explanation on the sharp peak of MHCI, CD4, CD8 expression showed in Fig. 3 > not done.

3.- Test LBBV vaccinated fish for LBBV RNA at times later than 7 days post-vaccination > not done.

4.- In the serial in vivo passage (Fig. 6): test samples from passages after the first one for LBBV on cell culture > not done.

Author Response

1.- Provide some data on fish antibody response to the vaccine > cannot be done (don´t have the anti-IgM antibody).

Reply: Thanks for your suggestion. In the future study, we will detect the antibody post vaccination.

2.- Provide an explanation on the sharp peak of MHCI, CD4, CD8 expression showed in Fig. 3 > not done.

Reply: Sorry, we did not get your question before. In this study, expressions of MHC I, CD8 and CD4 showed a sharp peak at 1 day post-vaccination, we speculated that the expression modle maybe relevant with rapid virus proliferation dynamics in host or disease onset speed. In our previosu paper [1], fish began to die at 1st day post LBBV challenge and the cumulative mortality reached 100% at 3 dpi by IP (Fig.1). And the LBBV replication in CPB cells showed the number of LBBV copies in CPB cells increased significantly from 10~12 h (Fig.2). So, LBBV avirulent strain infection by IP rapidly activated expressions of MHC I, CD8 and CD4 at 1 dpv.

[1] Determination and Characterization of a Novel Birnavirus Associated with Massive Mortality in Largemouth Bass. Microbiology Spectrum, 2022, 10(2): 10.1128/spectrum.01716-21.

3.- Test LBBV vaccinated fish for LBBV RNA at times later than 7 days post-vaccination > not done.

Reply: Sorry, we did not get your question before. Although we did not detect the LBBV in vaccinated fish later than 7 dpv, but we continously detected LBBV for 5 passages in virulence reversion test for vaccine safty evaluation. Because every passage was 4 days, the 5th passage is equivalent to the 20th day post vaccination. Our results showed no LBBV was detected by RT-PCR and cell culture test.

4.- In the serial in vivo passage (Fig. 6): test samples from passages after the first one for LBBV on cell culture > not done.

Reply: Sorry, we did not get your question before. qPCR is too sensitive because of detecting very short sequence compared to PCR, which leading to false positive diagnosis. Thus, we selected RT-PCR to detect the samples. Indeed, every passage samples were detected the LBBV by RT-PCR and cultured on cells according to《Announcement No.683 of the Ministry of Agriculture of the People’s Republic of China》. 

We have revised the method as follows: Reversion to virulence test was performed to evaluate the safety of avirulent vaccine. Largemouth bass were intraperitoneally injected with 0.05 mL avirulent vaccine at a dose of 107.5 TCID50 per fish or L-15 medium as a control. Fifteen fish were used for the every passage test, and 5 fish were randomly sampled at 4 dpi for the next passage injection. A part of the sampled spleen and kidney were subjected to the virus detection using cell culure on CPB and RT-PCR constructed in our lab [11]. Then the other part of the sampled organs was homogenized on ice with L-15 medium. The homogenate was centrifuged at 5000 rpm at 4 â—¦C for 15 min, then passed through a 0.22 μm pore size membrane filter (Millex HV; Millipore), and supernants was used for the next injection in fish. Subsequently, fifteen largemouth bass were intraperitoneally injected with 0.05 mL viral supernant and reared in the tanks. The injection step was repeated four times for the avirulent vaccine candidate passage in largemouth. At the last passge, five fish was sampled for LBBV detection by RT-PCR and cell culture, and the rest ten fish were reared for 21 days for abnormality and mortality observation.

Results was also added one senctence as follows:no live virus was isolated from passage 2 to passage 5”.

Round 3

Reviewer 2 Report

Comments and Suggestions for Authors

No further comments